# Obstetric risk profiles and causes of death: Estimating their association with cesarean sections among maternal deaths in Mexico

Pablo Martínez-Garrido[1], Jimena Fritz[1]*, Alejandra Montoya[2], Mayra J. Garza[1], Héctor Lamadrid-Figueroa[1]

1 Department of Perinatal Health, National Institute of Public Health, Cuernavaca, Morelos, Mexico, 2 Carlos Slim Foundation, Miguel Hidalgo, Mexico City, México

* jimenafritz.insp@gmail.com

**Data Availability Statement:** The data that support the findings of this study are openly available in

## Abstract

### Background

Maternal mortality is a critical indicator of healthcare quality, and in Mexico, this has become increasingly concerning due to the stagnation in its decline, alongside a concurrent increase in cesarean section (C-section) rates. This study characterizes maternal deaths in Mexico, focusing on estimating the association between obstetric risk profiles, cause of death, and mode of delivery.

### Methods

Utilizing a retrospective observational design, 4,561 maternal deaths in Mexico from 2010–2014 were analyzed. Data were sourced from the Deliberate Search and Reclassification of Maternal Deaths database, alongside other national databases. An algorithm was developed to extract the Robson Ten Group Classification System from clinical summaries text, facilitating a nuanced analysis of C-section rates. Information on the reasons for the performance of a C-section was also obtained. Logistic regression and multinomial logistic regression models were used to estimate the relation between obstetric risk factors, mode of delivery and causes of maternal death, adjusting for covariates.

### Results

Among maternal deaths in Mexico from 2010–2014, 47.1% underwent a C-section, with a significant history of previous C-sections observed in 31.4% of these cases, compared to 17.4% for vaginal deliveries (p<0.001). Early prenatal care in the first trimester was more common in C-section cases (46.8%) than in vaginal deliveries (38.3%, p<0.001). A stark contrast was noted in the place of death, with 82.4% of post-C-section deaths occurring in public institutions versus 69.1% following vaginal births. According to Robson's classification, the highest C-section rates were in Group 4 (67.2%, p<0.001) and Group 8 (66.9%, p<0.001). Logistic regression analysis revealed no significant difference in the odds of receiving a C-section in private versus other settings after adjusting for Robson criteria (OR:

figshare at https://doi.org/10.6084/m9.figshare.25587417.v1

**Funding:** The author(s) received no specific funding for this work.

**Competing interests:** The authors have declared that no competing interests exist.

1.21; 95% CI: 0.92, 1.60). A prior C-section significantly increased the likelihood of another (OR: 2.38; CI 95%: 2.01, 2.81). The analysis also indicated C-sections were significantly tied to deaths from hypertensive disorders (RRR = 1.25, 95% CI [1.12, 1.40]). In terms of indications, 6.3% of C-sections were performed under inadequate indications, while the indication was not identifiable in 33.1% of all C-sections.

## Conclusions

This study highlights a significant overuse of C-sections among maternal deaths in Mexico (2010–2014), revealed through the Robson classification and ana analysis of the reported indications for the procedure. It underscores the need for revising clinical decision-making to promote evidence-based guidelines and favor vaginal deliveries when possible. High C-section rates, especially noted disparities between private and public sectors, suggest economic and non-clinical factors may be at play. The importance of accurate data systems and further research with control groups to understand C-section practices' impact on maternal health is emphasized.

## Introduction

Maternal mortality is a significant public health concern, reflecting the overall quality of a healthcare system. In Mexico, significant efforts have been made to reduce maternal deaths, leading to a considerable decline over the past few decades. However, recent data indicate a slowing in the rate of decrease in maternal mortality. According to the Global Burden of Disease Study 2019, the MMR in Mexico decreased 15% from 1990 to 2000 (71.4 to 61.7 deaths per 100,000 live births); 14% from 2000 to 2010 (61.7 to 52.9) but only 8% from 2010 to 2019 (52.9 to 48.6), raising questions about potential contributing factors [1].

During the same period, there has been a continuous increase in C-section (C-section) rates in both public and private healthcare sectors in Mexico. The percentage of institutional births attended within the Mexican health system rose from 87.6% in 2000 to 95.1% in 2012, while the C-section rate rose from 28.8 per 100 births in 2000 to 45.1 per 100 births in 2015 [2].

This parallel trend between a slowing decrease in maternal mortality and rising cesarean rates prompts a critical examination. There is growing concern about potential overexposure of women to excessive and unnecessary cesarean procedures, which may contribute to complications for both mothers and newborns [3]. These complications include an increased risk of bleeding, infection, removal of the uterus, and injury to adjacent organs for mothers, and higher risks of respiratory complications for newborns [4].

A debate has emerged on the appropriate nationwide percentage of cesarean deliveries, the potential risks to both mothers and fetuses, and necessary policy interventions to reduce rates [5,6]. However, merely monitoring C-section rates is insufficient for understanding the problem and its causes. Amongst several potential strategies, a comprehensive analysis focusing on maternal deaths could provide valuable insights.

The objective of this study is to describe women who succumbed to maternal death, the circumstances around their death, and specifically to estimate associations between C-section rates, obstetric risk profiles, and other covariates, while ascertaining the extent to which C-sections were not adequately justified. While this work does not aim to establish causal relationships, this study contributes to the broader discussion on whether a substantial number of C-

sections leading up to maternal deaths may have been unnecessary and potentially contributed to the fatal outcomes.

## Methods

### Ethics statement

From an ethical standpoint, this project was classified as minimal-risk research, as it involved the analysis of de-identified administrative records on maternal deaths and births routinely generated by the Mexican General Directorate of Health Information. The study adhered to the principles outlined in the Declaration of Helsinki and was conducted with strict measures to ensure the privacy and confidentiality of the data subjects. Access to the data was granted under protocols designed to protect personal information, and the study received ethical approval from the Ethics Committee of the National Institute of Public Health of Mexico (authorization number CI: 1340).

### Study design, population, and data sources

This research was a retrospective observational study that analyzed 4,561 maternal deaths that occurred in Mexico between 2010 and 2014. The data for the study were obtained from the Deliberate Search and Reclassification of Maternal Deaths database (BIRMM, acronym in Spanish), generated by the General Directorate of Health Information of the Mexican Ministry of Health. Data were accessed and downloaded for research purposes on February 4th, 2015. The BIRMM process involved reviewing documentation and confirming maternal death cases based on analyses conducted by technical committees, which collected data from available sources, including death certificates, medical records, verbal autopsy records, and confidential enquiry or autopsy reports [7]. One output of this process was a short clinical summary or case history in text form, typically between 50 and 100 words, which included clinical information on the case. In addition to the maternal death data, the study incorporated information from other sources such as the marginalization and human development indices at the municipality level, provided by the National Population Council [8]. The analyzed data are routinely collected for health information systems and do not carry any identifiers.

### Variables

The study considered various individual-level characteristics. These variables included the mode of birth (vaginal birth vs. C-section), the mother's age, schooling, and access to healthcare. The mother's age was categorized into decades. Schooling was grouped into two categories: those with less than high school education and those who completed high school or pursued higher education. Access to healthcare was classified by the place where death occurred, which was categorized in the main public healthcare institutions of the country such as IMSS (Mexican Social Security Institute), ISSSTE (Social security and services Institute for State Workers), SSA/SP (Ministry of Health), or others, the private sector, or whether death occurred outside the health system. Basic causes of death were grouped into nine mutually exclusive categories as defined by the ICD-10: abortion; hypertensive disorders of pregnancy, delivery, and puerperium; obstetric hemorrhage; pregnancy-related infections; other obstetric complications; unforeseen treatment complications; non-obstetric complications, undetermined; other codes of interest, and contributing conditions [9]. Additionally, we characterized the immediate causes of death, as listed in the death certificates, by maternal death cause and delivery mode.

Lastly, we ascertained the presence of co-morbid conditions, specifically noncommunicable diseases such as diabetes, chronic hypertension, cancer, and autoimmune diseases, creating a dummy variable indicating mention of at least one of these conditions in any of the causes listed by the women's death certificates.

## Robson classification and algorithm construction

The Robson classification, or Ten Group Classification System (TGCS), is a system developed to categorize all women admitted for delivery into one of ten groups based on easily identifiable obstetric characteristics such as parity, gestational age, onset of labor, fetal presentation, and number of fetuses. This classification provides a standardized way to analyze and compare C-section rates across different populations and settings, facilitating the identification of patterns and trends, Table 1.

In this study, an algorithm was developed to extract relevant information from clinical summaries of each case, related to pregnancy and birth characteristics, allowing for the application of the Robson classification. The algorithm utilizes automated text coding techniques and regular expressions, using the *regexs* function of Stata v 17 (Stata, College Station, TX) [10]. This approach enables the identification and categorization of cases according to the Robson classification, even when variables of interest are not explicitly reported in standardized formats. In addition, we examined the clinical summaries to ascertain the indication for C-section in women who underwent the procedure, classifying it as Absolute, Relative, Inadequate or Not-recorded, in accordance with criteria proposed by Aranda-Neri et al. [11]. The latter classification considers the lack of recording of an indication for the C-section, as equivalent to an "inadequate" indication.

After estimating univariate statistics of the study variables, differences between groups defined by delivery mode, were analyzed by Student's t-test or Chi-square test for continuous and categorical variables, respectively.

A bivariate logistic regression model was fitted to estimate the probability of a C-section being performed conditional on Robson's risk categories. Additionally, a multiple logistic regression model was employed to model the performance of C-section in women who died,

**Table 1. The robson classification system, also known as the Ten Group Classification System (TGCS).**

| GROUP | GROUP CHARACTERISTICS |
|---|---|
| 1 | Nulliparous with a single fetus in cephalic presentation, 37 weeks or more of pregnancy, spontaneous labor |
| 2 | Nulliparous with a single fetus in cephalic presentation, 37 weeks or more of pregnancy, undergoing induction or cesarean before the start of labor |
| 3 | Multiparous without previous C-section, single fetus in cephalic presentation, 37 weeks or more of pregnancy, spontaneous labor |
| 4 | Multiparous without previous C-section, single fetus in cephalic presentation, 37 weeks or more of pregnancy, undergoing induction or cesarean before the start of labor |
| 5 | Multiparous with at least one previous cesarean, single fetus in cephalic presentation, 37 weeks or more of pregnancy |
| 6 | Nulliparous with a single fetus in breech presentation |
| 7 | Multiparous with a single fetus in breech presentation, including those with previous C-section |
| 8 | All women with multiple pregnancy, including those with previous cesarean |
| 9 | All women with a single fetus in a transverse or oblique position, including those with a previous caesarean section |
| 10 | All women with a single fetus in cephalic presentation of less than 37 weeks of pregnancy, including those with previous C-section |

adjusting for Robson's obstetric risk categories, place of death, access to social security, level of education, and the marginalization index of the municipality of residence.

In addition to the primary analyses, we fitted a multinomial logistic regression model to examine the relationship between the mode of birth and the specific cause of maternal death. The outcome variable in this model was the cause of death, while the main predictor was the mode of birth—either C-section or vaginal delivery. We adjusted the model for a range of covariates, including Robson's classification variables, place of death, access to social security, level of education, and the marginalization index of the municipality of residence. The multinomial logistic regression model was utilized to estimate the predicted probabilities of dying from each cause, by delivery mode, accounting for the influence of the covariates.

## Results

Table 2 presents the descriptive statistics along with the results from bivariate hypothesis testing. The analysis revealed that among the women who died from maternal causes during the study period, 47.1% had undergone a C-section. Furthermore, 31.4% of the women who underwent a C-section had a previous cesarean in a prior pregnancy. In contrast, only 17.4% of women who did not undergo a C-section had a previous cesarean in their medical history. This difference was statistically significant (p < 0.001).

In the study, among the women who died from maternal causes, 46.8% of those who underwent a C-section had their first prenatal visit in the first trimester, compared to 38.3% of women who did not undergo a C-section (p<0.001). Breech presentation was reported in 0.09% of C-sections and 0.04% in vaginal deliveries, with this difference not being statistically significant (p = 0.490).

Regarding comorbidities, 12.2% of the deceased who underwent a C-section and 12.6% of those who did not undergo the procedure were found to have chronic diseases, showing no significant difference between the two groups (p = 0.738).

The place of death analysis showed that 82.4% of the deaths following a C-section occurred in public institutions, compared to 69.1% for those who did not undergo the procedure, demonstrating a significant disparity (p<0.001). Additionally, C-sections were attended by medical personnel in 92.5% of cases, whereas vaginal births were attended by nurses or interns in 51.9%, by midwives in 9.1%, and by other persons in 5.7%, indicating significant differences in the type of delivery care provided (p<0.001).

The distribution of the sample according to the Robson classification, stratified by type of birth, is shown in Table 3. Group 4 (multiparous, without previous C-section, single fetus, cephalic presentation, ≥37 WOG, undergoing induction or cesarean) had the highest proportion of C-sections at 67.2% (p<0.001), followed by Group 8 (multiple pregnancy) with a 66.9% (p<0.001). In Group 9 (single fetus, transverse, or oblique position), a proportion of 64.7% of C-sections was observed, followed by Group 2 (nulliparous, single fetus, cephalic presentation, ≥37 WOG, undergoing induction or cesarean before starting labor) with a 57.89% of C-sections, although this difference was not statistically significant (p = 0.146 and p = 0.182, respectively). In group 10 (single fetus, cephalic presentation, ≥37 WOG) the proportion of C-sections was 49.2% (p = 0.29). Group 1 (nulliparous, single cephalic fetus, ≥37 weeks of gestation, spontaneous delivery), had a 48.9% percentage of C-sections, but this difference was not statistically significant (p = 0.248). Interestingly, in Group 5 (multiparous with at least one previous C-section, single fetus, cephalic presentation, and ≥37 weeks of gestation), where 59 women were classified, no C-section was performed.

By adjusting the logistic regression model with the Robson variables and cause of maternal death, the following was observed (Table 4). Women treated in private institutions had higher

**Table 2. Descriptive characteristics of maternal deaths in Mexico 2010–2014, stratified by birth mode [£].**

| | All maternal deaths (n = 4561) | No C-section (n = 2412) | C-section (n = 2149) | P-value ¥ |
|---|---|---|---|---|
| | | 52.88% | 47.11% | |
| | Mean ± SD or N (%) | | | |
| **Sociodemographic characteristics** | | | | |
| Age | 28.15±7.13 | 27.84±7.19 | 28.51±7.05 | <0.001 |
| **Marital status** | | | | <0.001 |
| Single | 3695(81.01%) | 1915(79.39%) | 1780(82.83%) | |
| Married/co-habiting | 756(16.58%) | 419(17.37%) | 337(15.68%) | |
| No data | 110(2.41%) | 78(3.23%) | 32(1.49%) | |
| **Schooling** | | | | <0.001 |
| < High school | 3150(69.06%) | 1756(72.8%) | 1394(64.87%) | |
| High school or more | 1133(24.84%) | 517(21.43%) | 616(28.66%) | |
| No data | 278(6.1%) | 139(5.76%) | 139(6.47%) | |
| **Occupation** | | | | <0.001 |
| Employed | 3263(71.54%) | 1751(72.6%) | 1512(70.36%) | |
| Unemployed | 147(3.22%) | 60(2.49%) | 87(4.05%) | |
| No data | 1151(25.24%) | 601(24.92%) | 550(25.59%) | |
| **Social security** | | | | <0.001 |
| Without social security: (SP or IMSS oportunidades) | 3131(68.65%) | 1714(71.06%) | 1417(65.94%) | |
| With social security (IMSS, ISSSTE, other) | 1121(24.58%) | 501(20.77%) | 620(28.85%) | |
| No data | 309(6.77%) | 197(8.17%) | 112(5.21%) | |
| **Obstetric history** | | | | |
| **Number of pregnancies** | | | | <0.001 |
| 1 | 1561(34.22%) | 789(32.71%) | 772(35.92%) | |
| 2 | 1754(38.46%) | 876(36.32%) | 878(40.86%) | |
| 3 or more | 1245(27.3%) | 746(30.93%) | 499(23.22%) | |
| Not specified | 1(0.02%) | 1(0.04%) | 0(0%) | |
| **Previous C-section s** | | | | <0.001 |
| No | 3450(75.64%) | 1982(82.17%) | 1468(68.31%) | |
| Yes | 1094(23.99%) | 419(17.37%) | 675(31.41%) | |
| Not specified | 17(0.37%) | 11(0.46%) | 6(0.28%) | |
| **Characteristics of last pregnancy** | | | | |
| 1[st] Prenatal Care Visit | | | | <0.001 |
| No prenatal care | 422(9.25%) | 285(11.82%) | 137(6.38%) | |
| In 1st trimester | 1930(42.32%) | 925(38.35%) | 1005(46.77%) | |
| In 2nd trimester | 789(17.3%) | 422(17.5%) | 367(17.08%) | |
| In 3rd trimester | 130(2.85%) | 77(3.19%) | 53(2.47%) | |
| Not registered | 1290(28.28%) | 703(29.15%) | 587(27.32%) | |
| Gestation weeks | 32.45±7.5 | 30.3±8.97 | 34.38±5.15 | <0.001 |
| <37 | 1751(59.02%) | 869(61.85%) | 882(56.47%) | |
| ≥37 | 1216(40.98%) | 536(38.15%) | 680(43.53%) | |
| **Breech presentation** | | | | 0.490 |
| No | 4558(99.93%) | 2411(99.96%) | 2147(99.91%) | |
| Yes | 3(0.07%) | 1(0.04%) | 2(0.09%) | |
| **Presence of co-morbidities*** | | | | 0.738 |
| No | 3996(87.61%) | 2109 (87.44%) | 1887 (87.81%) | |

*(Continued)*

**Table 2.** (Continued)

| | All maternal deaths (n = 4561) | No C-section (n = 2412) | C-section (n = 2149) | P-value ¥ |
|---|---|---|---|---|
| | | **52.88%** | **47.11%** | |
| Yes | 565 (12.39%) | 303 (12.56%) | 262 (12.19%) | |
| **Characteristics of death** | | | | |
| **Death site** | | | | <0.001 |
| Public institution | 3437(75.36%) | 1666(69.07%) | 1771(82.41%) | |
| Private institution | 341(7.48%) | 140(5.8%) | 201(9.35%) | |
| Outside the system | 735(16.11%) | 574(23.8%) | 161(7.49%) | |
| Not registered | 48(1.05%) | 32(1.33%) | 16(0.74%) | |
| **Delivery care** | | | | <0.001 |
| Physician, nurse or intern | 3240(71.04%) | 1253(51.95%) | 1987(92.46%) | |
| Midwife | 220(4.82%) | 220(9.12%) | 0(0%) | |
| Other | 140(3.07%) | 137(5.68%) | 3(0.14%) | |
| There was no delivery / abortion | 507(11.12%) | 507(21.02%) | 0(0%) | |
| Not registered | 454(9.95%) | 295(12.23%) | 159(7.4%) | |

¥ Student's T tests were performed for numerical variables (age) and Chi-square tests for categorical variables.

\* Defined as mention of diabetes, chronic hypertension, cancer, or autoimmune diseases in any cause listed in the death certificate.

odds of receiving a C-section compared to those treated in other settings section (OR = 1.21; 95% CI: 0.92, 1.60). Although the association is not statistically significant, it suggests a trend towards higher cesarean rates in private institutions. Women who died outside the healthcare system had the lowest odds of receiving a C-section (OR = 0.26; 95% CI: 0.20, 0.32).

Based on the model, after adjustment for covariates, the estimated proportions of women who underwent a C-section was highest in private institutions (0.60, CI 95%: 0.54–0.65). IMSS showed the second highest proportion (0.55, 95% CI: 0.51, 0.58), followed by other public institutions such as SEDENA (Army), SEMAR (Navy), and PEMEX (Mexican Petroleum) (0.53, 95% CI: 0.47, 0.58) and ISSSTE (0.53, 95% CI: 0.42, 0.63).

### Indications for the performance of C-section

Overall, 50.1% of C-sections were performed after an absolute indication of the procedure, while 10.3% were performed under a relative indication. In 6.3% of the cases, an inadequate indication for the C-section was recorded, while no reason for indicating a C-section was recorded in 33.1% of the cases.

Breaking down the adequacy of the C-section indication by Robson risk categories yielded the following results: In risk group 1 (Nulliparous with a single fetus in cephalic presentation, 37 weeks or more of pregnancy, spontaneous labor) 40% of C-sections were either justified on a wrong indication or the reasons for the indication were not recorded. In group 2 (Nulliparous with a single fetus in cephalic presentation, 37 weeks or more of pregnancy, undergoing induction or cesarean before the start of labor) this figure rose to 54% and only 9% had absolute indication for a C-section. In group 3 (Multiparous without previous C-section, single fetus in cephalic presentation, 37 weeks or more of pregnancy, spontaneous labor) the majority (over 53%) had an absolute indication for C-section, while 36% were unjustified or the motive was unknown. In group 4 (Multiparous without previous C-section, single fetus in cephalic presentation, 37 weeks or more of pregnancy, undergoing induction or cesarean before the

**Table 3. Distribution of maternal deaths by Robson's classifications group * and type of birth.**

| Group | Vaginal delivery | C-section | Total | p-value ** |
|-------|------------------|-----------|-------|-----------|
| **1** | 440 | 421 | 861 | |
| | 51.10% | 48.90% | 100% | 0.248 |
| **2** | 16 | 22 | 38 | |
| | 42.11% | 57.89% | 100% | 0.182 |
| **3** | 933 | 687 | 1620 | |
| | 57.59% | 42.41% | 100% | <0.001 |
| **4** | 40 | 82 | 122 | |
| | 32.79% | 67.21% | 100% | <0.001 |
| **5** | 59 | 0 | 59 | |
| | 100% | 0% | 100% | <0.001 |
| **6** | 0 | 1 | 1 | |
| | 0.00% | 100.00% | 100% | 0.289 |
| **7** | 1 | 1 | 2 | |
| | 50.00% | 50.00% | 100% | 0.935 |
| **8** | 35 | 71 | 106 | |
| | 33.02% | 66.98% | 100% | <0.001 |
| **9** | 6 | 11 | 17 | |
| | 35.29% | 64.71% | 100% | 0.146 |
| **10** | 881 | 853 | 1734 | |
| | 50.81% | 49.19% | 100% | 0.029 |
| **Total** | 2411 | 2149 | 4560 | |
| | 52.87% | 47.13% | 100% | |

* According to Robson's classification.

start of labor), 23.2% had a relative indication of C-section, while 34% had an inadequate or unknown indication for the procedure. Group 8 (All women with multiple pregnancy, including those with previous cesarean) had the highest percentage of relative indications with 32%. Finally, in group 10 (All women with a single fetus in cephalic presentation of less than 37 weeks of pregnancy, including those with previous C-section), most women (51%) had an absolute indication for C-section, while 43% had an inadequate or unspecified justification. Other risk categories had very small numbers. Table 5 fully discloses the distribution of indications (absolute, relative, inadequate, and unspecified) by Robson's risk group. Three cases did not have enough information for classification.

## Cause of death

The most frequent immediate causes of death registered in the death certificate, by maternal death cause, are presented in S1 and S2 Tables. For women that did not undergo a C-section, and other than "unspecified symptoms", intracerebral hemorrhage was the leading immediate cause of death in women who died of hypertensive disorders of pregnancy. Septicemia was the leading immediate cause for women who were classified as having died of pregnancy-related infections, contributory conditions, indirect deaths, and "others". Pulmonary embolism was the leading cause in women with other obstetric complications. Unspecified shock was the leading cause of death amongst women who died of obstetric hemorrhage, and anaphylactic shock was the leading cause for those whose death was attributed to complications of management. Women who had a C-section had a similar profile of immediate causes of death, except

**Table 4. Odds Ratios of birth by C-section in maternal Deaths in Mexico 2010–2014.** Logistic regression model adjusted for age, schooling, marital status, previous pregnancies, prenatal control, site of death, marginalization, and human development index of municipality of residence.

| C-section | OR* | P Value | (95% Confidence interval) | |
|---|---|---|---|---|
| **Death site** | | | | |
| ISSSTE | 0.921 | 0.717 | 0.590 | 1.438 |
| SSa | 0.817 | 0.023 | 0.686 | 0.973 |
| Other public | 0.923 | 0.567 | 0.703 | 1.213 |
| Private institution | 1.217 | 0.167 | 0.922 | 1.606 |
| Outside the system | 0.257 | < 0.001 | 0.202 | 0.326 |
| **Cause of maternal death** | | | | |
| Obstetric hemorrhage | 0.273 | < 0.001 | 0.223 | 0.334 |
| Infection related to pregnancy | 0.316 | < 0.001 | 0.219 | 0.455 |
| Other obstetric complications | 0.401 | < 0.001 | 0.312 | 0.516 |
| Unanticipated complications of labor management | 0.935 | 0.864 | 0.431 | 2.026 |
| Non-obstetric complications | 0.260 | < 0.001 | 0.217 | 0.312 |
| Unknown / Undetermined | 0.775 | 0.694 | 0.218 | 2.755 |
| Other codes of interest | 0.420 | < 0.001 | 0.328 | 0.538 |
| Contributory conditions | 0.183 | < 0.001 | 0.126 | 0.265 |
| **Robson variables** | | | | |
| Multiparous | 0.659 | < 0.001 | 0.569 | 0.764 |
| Previous C-Section | 2.383 | < 0.001 | 2.019 | 2.812 |
| Multiple pregnancy | 2.160 | < 0.001 | 1.370 | 3.406 |
| Weeks of gestation | 0.936 | 0.352 | 0.815 | 1.076 |
| Induction | 0.584 | 0.004 | 0.404 | 0.845 |
| Presentation | 2.528 | 0.463 | 0.212 | 30.113 |
| Position | 2.658 | 0.059 | 0.965 | 7.326 |
| Scheduled for C-section | 6.847 | < 0.001 | 3.958 | 11.846 |

*OR: Odds ratio. When examining the Robson variables, the following associations were observed: Women with a fetus in the transverse position had a higher likelihood of receiving a C-section (OR = 2.65; CI 95%: 0.96, 7.32). Although the association is not statistically significant, it suggests an increased risk of cesarean delivery for women with transverse fetal position. Women with a fetus in breech presentation also had a higher likelihood of receiving a C-section (OR = 2.52; CI 95%: 0.21, 30.11). However, due to the wide confidence interval, the association is not statistically significant, indicating uncertainty in the estimate. Women with a previous C-section had a significantly higher probability of receiving a C-section (OR = 2.38; CI 95%: 2.01, 2.81). Women with multiple pregnancies also had a significantly higher likelihood of receiving a C-section (OR = 2.16; CI 95%: 1.37, 3.41).

for septicemia being the leading cause of immediate death on those women who died of complications of management.

In the multinomial logistic regression analysis, various predictors were significantly associated with different competing causes of maternal death, relative to the base outcome of 'non-obstetric complications'. The mode of birth (C-section) was significantly associated with death due to hypertensive disorders (RRR = 1.25, 95% CI [1.12, 1.40]). Women with two or more previous deliveries had a higher likelihood of death from obstetric hemorrhage (RRR = 1.36, 95% CI [1.20, 1.54]). Similarly, a history of previous C-sections showed a significant association with death from other obstetric complications (RRR = 1.18, 95% CI [1.03, 1.36]). Multiple pregnancies were notably related to death from obstetric hemorrhage (RRR = 2.47, 95% CI [1.90, 3.21]). Gestational age was associated with death from pregnancy-related infections (RRR = 1.45, 95% CI [1.18, 1.79]). Labor induction was linked to death from hypertensive

**Table 5. Classification of C-sections according to adequacy of surgical indication.**

| Robson's Risk Category | Absolute | Relative | Inadequate | Not Recorded (NA) | Row Total |
|---|---|---|---|---|---|
| 1 | 201 (47.74%) | 51 (12.11%) | 37 (8.79%) | 132 (31.35%) | 421 (100%) |
| 2 | 2 (9.09%) | 8 (36.36%) | 4 (18.18%) | 8 (36.36%) | 22 (100%) |
| 3 | 368 (53.64%) | 66 (9.62%) | 37 (5.39%) | 215 (31.34%) | 686 (100%) |
| 4 | 35 (42.68%) | 19 (23.17%) | 8 (9.76%) | 20 (24.39%) | 82 (100%) |
| 5 | 0 (0.0%) | 0 (0.0%) | 0 (0.0%) | 0 (0.0%) | 0 (—) |
| 6 | 0 (0.0%) | 1 (100.0%) | 0 (0.0%) | 0 (0.0%) | 1 (100%) |
| 7 | 1 (100.0%) | 0 (0.0%) | 0 (0.0%) | 0 (0.0%) | 1 (100%) |
| 8 | 29 (40.85%) | 23 (32.39%) | 3 (4.23%) | 16 (22.54%) | 71 (100%) |
| 9 | 10 (90.91%) | 1 (9.09%) | 0 (0.0%) | 0 (0.0%) | 11 (100%) |
| 10 | 431 (50.65%) | 53 (6.23%) | 47 (5.52%) | 320 (37.6%) | 851 (100%) |
| Total | 1077 (50.19%) | 222 (10.34%) | 136 (6.34%) | 711 (33.13%) | 2146 (100%) |

disorders (RRR = 1.22, 95% CI [1.09, 1.38]). Lastly, a pre-scheduled C-section was associated with hypertensive disorders as the cause of death (RRR = 1.33, 95% CI [1.20, 1.49]).

In the examination of the correlation between the method of delivery and the cause of maternal death, several distinct patterns were observed (Fig 1). Women who underwent a C-section exhibited a significantly higher probability of death due to hypertensive disorders as compared to their counterparts who underwent vaginal deliveries. Conversely, hemorrhage as a cause of death was marginally more prevalent among women who had vaginal deliveries than those who had C-sections. While sepsis-related deaths showed a slightly increased tendency among the C-section group, the overlap in confidence intervals suggests that the difference might not be robustly significant.

For other categorically defined causes of death, there was a minor elevated risk for women who underwent vaginal deliveries. The probability of deaths associated with management issues was consistent between the two groups, albeit with a minute inclination towards the C-section group. Both methods of delivery, vaginal and C-section, presented nearly identical probabilities when it came to deaths from indirect causes. Deaths occurring late postpartum had a marginally higher likelihood in the vaginal delivery group, contrasting with deaths due to causes outside the chapter, which showed a pronounced higher risk in the vaginal delivery cohort.

## Discussion

Our study described and analyzed the characteristics of women who died of maternal causes, with a focus on those women who underwent a C-section and their justification, through data extracted from the BIRMM maternal mortality database for the period 2010–2014 in Mexico. To the best of our knowledge, no prior studies in our country have explored this aspect using this data source. Overall, the proportion of women who succumbed to maternal death after a C-section in Mexico during the study period was akin to figures in South Africa, another middle-income country with similarities to Mexico [12].

Stratification into Robson's groups is beneficial in identifying the obstetric and gynecological characteristics impacting C-section practices [13]. This classification system helps in distinguishing cases where the procedure was appropriately indicated from cases where it may have been unnecessary. Analyzing data within each group facilitates the identification of improvement areas and the development of strategies to reduce the high rates of C-sections in Mexico. This approach enables targeted interventions and interventions to ensure that C-sections are performed when medically necessary, thereby promoting better maternal and neonatal outcomes.

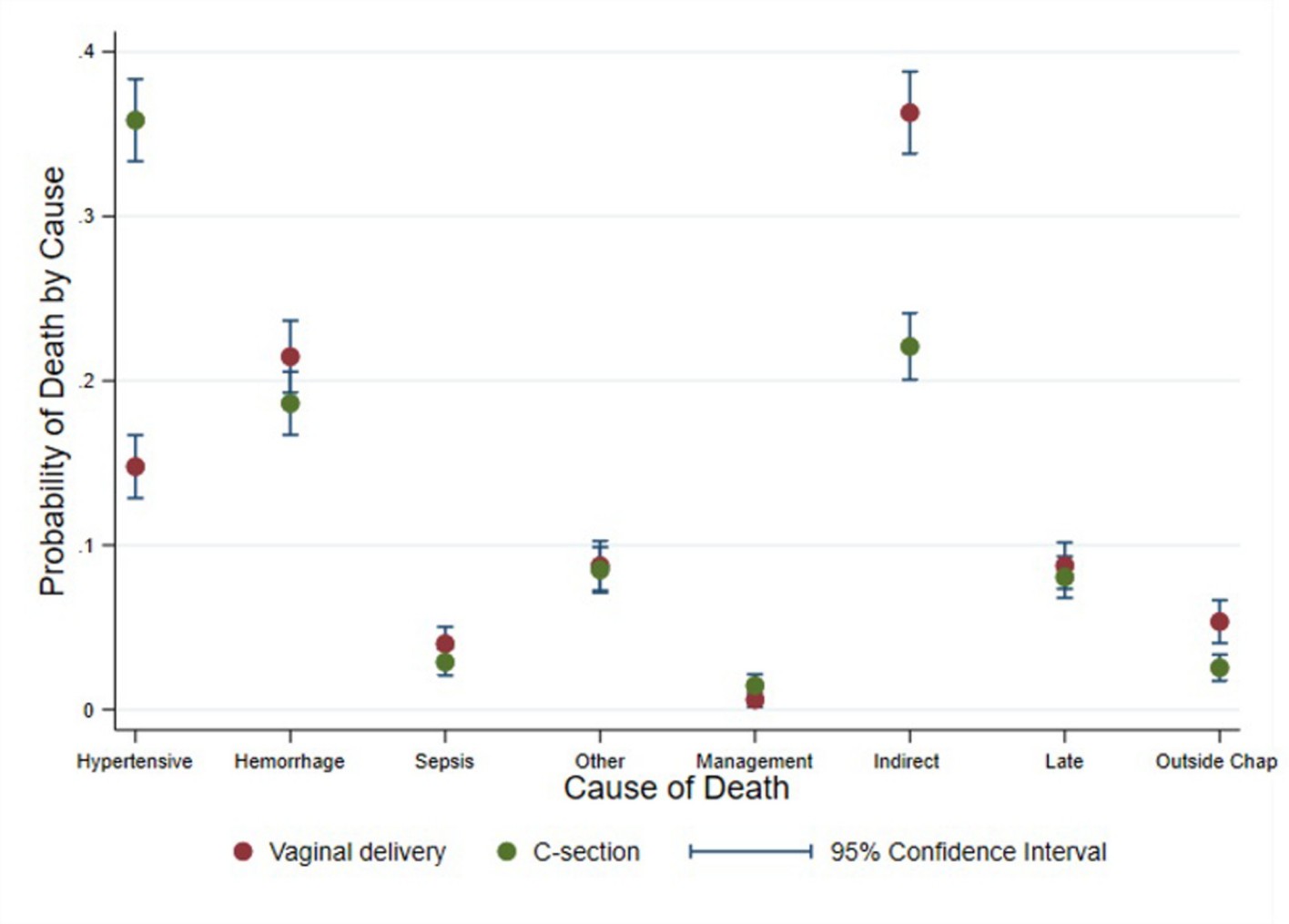

**Fig 1. Adjusted (multinomial logistic model) probability of death by obstetric cause group and mode of delivery.**

Recent literature supports the Robson classification as a valuable tool for assessing C-section practices. For instance, the World Health Organization recommends the use of the Robson Ten-Group Classification System (RTGCS) as an effective tool for monitoring and analyzing C-sections use [5]. Similarly, a study conducted at La Ribera University Hospital over nine years using the RTGCS highlighted its utility in assessing C-section rates [14].

Our findings suggest an overuse of C-sections based on the obstetric risk profile. In group 4 (multiparous women without a previous uterine scar, carrying a single pregnancy in cephalic position, at or beyond 37 WOG), a high proportion of C-sections (67.21%) was observed, despite this group not having any specific indications for a C-section unless complications arise during labor. In this group, only 42% of women had an absolute indication for C-section, while 9.8% had an inadequate indication listed as the reason for the intervention and in 24% no indication was listed in the clinical summary. Two of the four most frequent indications for C-section in this group are clearly inadequate: unspecified, previous C-section, the other two being placenta previa and lack of progression of labor. Looking at the causes of death amongst women in this group, obstetric hemorrhage was the leading cause, and C-section s were performed on 61% of these women. It is worth noting that in the case of multiparous women,

labor tends to favor spontaneous vaginal delivery. Therefore, the high rate of inadequate or missing indication for C-section in this group suggests an overuse of this surgical intervention, disregarding the favorable likelihood of spontaneous labor for multiparous women.

In deaths classified in group 10 (single fetus, cephalic presentation of $\geq$ 37 WOG, including those with previous cesarean), the proportion of C-sections was 49.19%. However, recent studies indicate that a previous cesarean is not an absolute indication for performing a C-section. Similarly, in group 3 (multiparous women without previous uterine scar, single pregnancy, cephalic presentation, $\geq$ 37 WOG, and spontaneous delivery), a proportion of 42.41% C-sections was observed and in group 1 (nulliparous women with single pregnancy, cephalic presentation, $\geq$ 37 WOG, and spontaneous delivery), the proportion of C-sections was 48.9%. The proportion of women in groups 3 and 10 that had an absolute indication for C-section were only slightly above 50%, and 47% in group 1, while blatantly inadequate indications in this group were reported in between 5 and 9% of the cases. These findings support the notion of an overuse of C-sections in Mexico, as these groups do not have inherent characteristics that necessitate a cesarean delivery.

It is interesting to note the disparity between the findings of the present study and the study conducted by Tura et al, where they reported that the largest proportion of C-sections was performed in groups 3, 5 and 1, with 21.4%, 21.1%, and 19.3% respectively [15]. Despite the fact that a direct comparison is not possible as our study only focused on women who died, the relatively large proportion of women in these groups in our study that had an inadequate or unspecified indication for the C-section is worth noticing. It is important to consider that healthcare practices and trends can vary across different settings and time periods, leading to variations in the proportions of C-sections among different Robson groups. Further research and analysis may be needed to better understand and compare the factors influencing C-section rates in different contexts.

Regarding prenatal control, 46.77% of women who underwent a C-section had their 1st prenatal control visit in the first trimester, while in vaginal deliveries this figure was 38.35%. Of women who underwent a C-section and had their first prenatal control visit in the 1st trimester, above 7% had an inadequate indication for the C-section, while the indication was unspecified (and thus potentially inadequate) in up to 32% of the cases. Inadequate indications explicitly stated in these cases were for example (four most frequent): ruptured membranes, fetal bradycardia, oligohydramnios, and nuchal cord. These figures indicate that, despite receiving presumably adequate prenatal care, some C-sections were performed without sufficient medical justification during labor and delivery. This highlights the need for a critical examination of the decision-making process surrounding C-sections and emphasizes the importance of ensuring that medical interventions are warranted and evidence based.

In addition to clinical aspects, research suggests that structural factors and the organization of obstetrics and gynecology services within each institution play a significant role in the decision-making process of performing a C-section. For example, it has been found that the odds of a pregnancy ending in a C-section are over 6-fold in Mexican private hospitals vs public institutions [11]. It has been found that deliveries attended during office hours pose a higher risk for C-section indication [16]. A qualitative study suggests that cesarean deliveries provide doctors with greater flexibility in scheduling without conflicts with their working hours [17]. Time-wise, there is a substantial difference between attending three cesarean deliveries compared to three vaginal deliveries. These factors indicate that non-clinical considerations, such as convenience and efficiency, along with maternal preferences, may influence the decision-making process regarding C-section, highlighting the need to address these structural factors to promote appropriate and evidence-based care during childbirth.

Economic incentives also play a significant role in the performance of C-sections, which could explain the very high C-section rates observed in the private sector in Mexico, not only in women who died but in all births [18]. It has been suggested that factors related to the disproportionate use of this procedure in the private sector are economic incentives at the health system, facility, and physician levels, such as insurance policies favoring cesarean coverage, profit-driven hospital practices, and higher physician reimbursement for cesareans [19]. Additionally, patient preferences, including fear of pain and concerns over labor progression and newborn well-being, contribute to higher cesarean rates [20]. Campero et al suggest that certain healthcare providers, driven by economic benefits and claiming greater safety and fewer side effects, have contributed to the demand for this surgical procedure among women from more privileged economic backgrounds [21]. This emphasizes the importance of addressing these economic factors and ensuring that medical decisions are based on clinical need and evidence-based practices rather than financial considerations.

Gender and years of experience of health personnel also contribute to the risk of C-section. A study in Mexico City found that younger obstetricians, are more likely to perform this procedure [22]. This could be attributed to the role of gynecologists in training medical interns. Additionally, the study suggests that female doctors tend to perform more C-sections than male obstetricians [22]. These findings indicate that the gender and experience level of healthcare providers can influence the decision-making process and contribute to the higher rates of C-sections.

The discussion of the findings regarding the association between the mode of delivery and causes of maternal death can be enriched by the insights from various studies. The identified patterns, especially the higher likelihood of death due to hypertensive disorders in women who had C-sections, align with existing literature. For instance, a study by Lisonkova et al. found that the risk of severe maternal morbidity was higher in planned cesarean deliveries compared to planned vaginal deliveries, particularly due to hypertensive disorders [23]. The latter study's focus on morbidity and the characteristics of its population makes it not necessarily comparable to our findings. The marginally higher prevalence of hemorrhage as a cause of death among women who had vaginal deliveries also resonates with the broad understanding of obstetric hemorrhage being a leading cause of maternal mortality globally. A study by Knight et al. in the UK, albeit in a different geographic context, highlights hemorrhage as a significant cause of maternal death [24]. A study in South Africa, identified hemorrhage as the leading cause of death after C-section [12]. These results are however in contrast to what we observed in Mexico, as the leading cause of death after C-section in our results were hypertensive disorders.

The slight increase in sepsis-related deaths among the C-section group could be related to the invasive nature of the procedure, which potentially exposes women to infection risks. In fact, looking at the immediate causes of death after C-section, septicemia was the leading immediate cause of death in women classified as having unanticipated complications of management; this in contrast to women who had a vaginal delivery, in which case the leading immediate cause after complications of management was anaphylactic shock. This is supported by a study by Kourtis et al. that associates cesarean delivery with a higher risk of postpartum infection compared to vaginal delivery [25]. The variances in causes of maternal death based on the mode of delivery stress the importance of continuous research and data analysis to further understand these associations and inform clinical practices to enhance maternal health outcomes.

The most notable limitation of this study is the exclusive focus on maternal deaths, without a comparative analysis involving women who underwent C-sections and survived. The absence of a control group of surviving women significantly hampers the ability to ascertain

causal relationships between C-section rates and maternal mortality. For a more robust understanding and to establish causality, future studies should include a control group of surviving women who underwent C-sections. The reliance on the BIRMM database, which was not explicitly designed for analyzing C-section practices, presents challenges in data completeness and accuracy. The lack of standardized criteria for filling out clinical summaries and the observed sub-reporting in medical abstracts could potentially introduce biases in the analysis. This limitation underscores the need for more precise and comprehensive data collection methods in future research.

The study was further constrained by the unavailability of data on the specific medical indications for performing C-sections. Without this crucial information, it's challenging to differentiate between necessary and potentially unnecessary C-sections, which is fundamental for a nuanced understanding of the observed trends. While the developed algorithm facilitated the classification of cases into Robson groups, its efficiency is bound to the quality and completeness of the data it processes. Any misclassification or oversight could affect the interpretation of findings. The study might not have accounted for all potential confounding factors that could influence both the likelihood of receiving a C-section and the risk of maternal death. These unobserved confounders might bias the associations observed in the study. The variability in healthcare practices across different geographical locations and institutions within Mexico might affect the generalizability of the findings. Differences in healthcare quality, medical training, and institutional policies could contribute to the observed variations in C-section rates and maternal mortality. Although we controlled the analysis for place of death, as way to consider variations in quality of care between different institutions, this information fails to capture nuances such as sudden events that led to deaths occurring before the patient reached medical care. The retrospective design of the study relies on historical data, which might not capture the current practices or trends in C-section rates and maternal mortality. While the study touched on socio-economic factors, a more in-depth examination of socio-economic data might provide additional insights into the disparities observed in C-section rates across different demographic groups. Addressing these limitations in future studies, particularly by incorporating a control group of surviving women and improving data collection methods, could significantly enhance the understanding of the complex dynamics between C-section practices and maternal mortality in Mexico.

## Conclusions

In conclusion, our study illuminates the patterns of cesarean section practices among maternal deaths in Mexico from 2010 to 2014, using the Robson classification as a framework. The findings reveal a concerning overuse of cesarean sections across various obstetric groups, particularly in scenarios where the obstetric risk profile does not warrant such interventions. This overuse, coupled with notable disparities in prenatal care between cesarean and vaginal deliveries, underscores the critical need to revisit and possibly revise the clinical decision-making processes surrounding cesarean sections within the Mexican healthcare system.

The observed high C-section rates among maternal deaths call for a more detailed examination of the indications leading to C-sections, with a special emphasis on enhancing the adherence to evidence-based guidelines and promoting vaginal delivery as a preferred, safer alternative whenever medically appropriate. The stark contrast in C-section rates between private and public institutions hints at underlying structural, economic, and possibly non-clinical factors influencing C-section decisions. Addressing these factors might require a multifaceted approach that encompasses policy adjustments, healthcare personnel education, improved patient-provider communication, and enhanced monitoring and evaluation of childbirth

practices. Our study also underscores the paramount importance of having comprehensive, standardized, and accurate data recording systems in place, not only to facilitate robust research but also to inform policy and practice aimed at optimizing maternal and neonatal health outcomes. The distinctive focus on maternal deaths in this study, while providing valuable insights, underscores the necessity of further research involving a control group, in order to find the causal relationships between C-section practices and maternal health outcomes comprehensively.

## Supporting information

**S1 Table. Ten most frequent immediate causes of maternal death amongst women who did not undergo a C-section in Mexico, 2010–2014.**
(DOCX)

**S2 Table. Ten most frequent immediate causes of maternal death amongst women who had a C-section in Mexico, 2010–2014.**
(DOCX)

## Author Contributions

**Conceptualization:** Pablo Martínez-Garrido, Jimena Fritz.

**Data curation:** Alejandra Montoya, Mayra J. Garza, Héctor Lamadrid-Figueroa.

**Formal analysis:** Alejandra Montoya, Héctor Lamadrid-Figueroa.

**Investigation:** Pablo Martínez-Garrido, Mayra J. Garza.

**Methodology:** Héctor Lamadrid-Figueroa.

**Supervision:** Jimena Fritz.

**Writing – original draft:** Pablo Martínez-Garrido.

**Writing – review & editing:** Jimena Fritz, Alejandra Montoya, Mayra J. Garza, Héctor Lamadrid-Figueroa.

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
