## [Decision Letter · Decision Letter 0]

2 Jan 2024

PONE-D-23-37685Cesarean sections among maternal deaths in Mexico: Obstetric Risk Profiles and Other CorrelatesPLOS ONE

Dear Dr. Fritz,

Thank you for submitting your manuscript to PLOS ONE. After careful consideration, we feel that it has merit but does not fully meet PLOS ONE’s publication criteria as it currently stands. Therefore, we invite you to submit a revised version of the manuscript that addresses the points raised during the review process. The revision outlined in the attachment  (which also addresses the reviewers comments) are required for us to consider the paper for publication. 

 I am confident that the issues raised can be addressed hence the recommendation for 'major revision' instead of outright rejection . Because of the work  required, I would like to suggest that you consider submitting the revised manuscript by Feb 16 2024 11:59PM. If you will need more time than this to complete your revisions, please reply to this message or contact the journal office at plosone@plos.org. Please include the following items when submitting your revised manuscript:A rebuttal letter that responds to each point raised by the academic editor and reviewer(s). You should upload this letter as a separate file labeled 'Response to Reviewers'.A marked-up copy of your manuscript that highlights changes made to the original version. You should upload this as a separate file labeled 'Revised Manuscript with Track Changes'.An unmarked version of your revised paper without tracked changes. You should upload this as a separate file labeled 'Manuscript'.

We look forward to receiving your revised manuscript.

Kind regards,

 Lawrence Chauke, PHD, MSc (Clinical Research), Cert Maternal and Fetal Medicine (SA), MMed (O&G), FCOG(SA), MBCHB, BTh, Dip HIV Man, 

Academic Editor

PLOS ONE

4. In the online submission form, you indicated that your data will be submitted to a repository upon acceptance.  We strongly recommend all authors deposit their data before acceptance, as the process can be lengthy and hold up publication timelines. Please note that, though access restrictions are acceptable now, your entire minimal  dataset will need to be made freely accessible if your manuscript is accepted for publication. This policy applies to all data except where public deposition would breach compliance with the protocol approved by your research ethics board. If you are unable to adhere to our open data policy, please kindly revise your statement to explain your reasoning and we will seek the editor's input on an exemption. 

6. Please ensure that you refer to Figure 1 in your text as, if accepted, production will need this reference to link the reader to the figure.

Additional Editor Comments:

Dear Fritz, Jimena and colleagues

Thank you very much to you and your team for submitting your research for consideration for publication to PLOSONE. The paper addresses and important topic of global interest. The rising rate of caesarean section and the less than optimal decline in maternal mortality is a major concern globally . Indeed existing data suggests that the world might not achieve the Sustainable Development Goal (SDG) of reducing maternal mortality to less than 70 000 per 100 000 livebirths by the year 2030 if the situation continue as is. Studies like yours is important in that they provide additional insight into some of the factors that should be looked into in the global pursuit to achieve the 2030 SDG 3.1. You provide a compelling background and justification for the study . Thank you for doing that. The paper is a bit difficult to follow as is at present and perhaps you should look at the following points to try and improve the readability of your work and also the chances of the paper being published:

1. Title

On reading the paper one gets the impression that the aim of the paper was to assess the association between mode of delivery, obstetrics risk factors and maternal deaths. If that is the case, you might want to revise the title to reflect this. Please refer to the PLOSONE author resource (https://plos.org/resource/how-to-write-a-great-title/) for guidance on how to write a winning title.

2.Research aim/objectives

Consider using scientific language to communicate the aim of the study and avoid using flowery words like unravel and elucidating. The reader should be very clear from reading the aim of the study as to what the study is all about. This is a bit not clear at present. May be the aim should be to assesses the association between the mode of delivery , maternal characteristics and maternal deaths or assess the contribution of mode of delivery and maternal characteristic on maternal deaths , for example.

3. Method

3.1. Thanks for describing the sources of your data

3.2. Consider writing this section in a past tense as recommended by the reviewers

3.3. Outcome variables: It is very important to chose the outcome variables carefully to ensure that they answer the research question/ objectives. The socio-demographic and pregnancy variables chosen are appropriate. It would have been more useful to also look at co-morbid conditions as these could potentially influence maternal deaths. In other words, is it simply having a caesarean section or having a caesarean section in the background of other co-morbid conditions (or other factors) that increases the risk of maternal death?. What about the experience of the surgeon and the level of care of the public healthcare facilities since majority of the deaths took place in public sector facilities. The following article might be of assistance ( Gebhardt G S, Fawcus S, Moodley J, Farina Z. Maternal death and caesarean section in South Africa: Results from the 2011 - 2013 Saving Mothers Report of the National Committee for Confidential Enquiries into Maternal Deaths. SAMJ, S. Afr. med. j. [Internet]. 2015 Apr [cited 2023 Dec 29] ; 105( 4 ): 287-291. Available from: http://www.scielo.org.za/scielo.php?script=sci_arttext&pid=S0256-95742015000300025&lng=en. http://dx.doi.org/10.7196/SAMJ.9351.)

4. Results

4.1. From the data, it seems like the were more deaths in the normal vaginal delivery group compared to caesarean section but you seem to lean more on caesarean section related maternal deaths which are less compared to the normal delivery group. There is also notable significant difference between the women who died after caesarean section and those following normal delivery but this data does not seem to be addressed by the paper. It might be worth spending some time looking and scrutinising this data as this might be something you should look include in your multimodal regression analysis.

4.2. Thank you for using the Robson classification to show the most indication for caesarean section and the risk of dying according to the Robson group.

4.3. Consider separating diagnosis and cause of death. Haemorrhage is the triggering event but what did the women die of ? Is it hypovolaemic shock? Those with pre-eclampsia did they die from bleeding , brain haemorrhage , embolism, etc? You tried to do this in Table 4 but can be further improved. Perhaps you need to scrutinise the data even further.

5. Conclusion: It would be appropriate to comment on this after the revised version because of the lack of alignment between the aim of the study, challenges with the interpretation and the conclusion arrived at.

6. Additional comments: Please see incorporate the specific comments from the reviewers in addition to the above.

I hope that the review does not discourage you. Looking forward to the revised version.

Reviewers' comments:

Reviewer's Responses to Questions

**Comments to the Author**

1. Is the manuscript technically sound, and do the data support the conclusions?

Reviewer #1: No

Reviewer #2: Yes

2. Has the statistical analysis been performed appropriately and rigorously? 

Reviewer #1: Yes

Reviewer #2: Yes

3. Have the authors made all data underlying the findings in their manuscript fully available?

Reviewer #1: No

Reviewer #2: Yes

4. Is the manuscript presented in an intelligible fashion and written in standard English?

Reviewer #1: No

Reviewer #2: Yes

5. Review Comments to the Author

Reviewer #1: General

Thank you for submitting the manuscript and the intention to share clinical experience and possible solutions for the Mexican populations with the rest of readership of PLOS ONE journals.

The title of the study “Cesarean sections among maternal deaths in Mexico: Obstetric Risk Profiles and

Other Correlates” suggest that maternal death will be a major point of discussion. However, this was not the case. Perhaps consider revising the title.

The manuscript requires Proofreading for scientific language and grammatical errors.

Introduction

Third paragraph about the trend of slowing in the decrease of maternal deaths…

Please such statistics should be made available here for the reader to understand that there has been a slowdown.

Fourth paragraph about potential strategies…

This should be noted that it cannot lead to understanding the cause as association does not equal casualty.

The last paragraph rationalizing the study….

How was the deduction of the caesarean sections being unnecessary and potentially contributing to maternal deaths made?

Study design, population, and data sources

This section should be written in past tense.

There are grammatical errors that need to be corrected. I recommend proofreading by an experienced proofreader or author.

Sentence that begins with “Briefly the BIRMM…”, need to be reconstructed. The word "briefly" appears to be related to what the BIRMM does. Perhaps remove it as it makes the sentence lose its meaning.

Use of Data and ethics: Even though such is readily accessible, permission to be used for research still need to be obtained. Therefore, this section should also indicate such. The use of retrospective data does not imply that ethics clearance is not required, kindly include this information on this section.

On variables: write in past tense.

A sentence on “years of schooling…”, this needs to be clarified. Was this about grades? It cannot be the number of years at school as that may also include those who had failed as progressing as the number of years, they are at school increases.

Section with “place where death occurred…”, events such as dying at home doesn’t always imply that there was no access to healthcare. Patients could have had a Pulmonary Embolus and die before transportation to hospital.

Statistical Analysis

Second paragraph on logistic regression: What was the rationale for estimating the probability of a Caesarean Section being done?

Results

There are several grammatical errors that need to be corrected.

Sentence “it was found that among the total number of births by c-section, 31.4% of women had a previous caesarean section, whereas in vaginal deliveries, only 17.4% had this characteristic (p<0.001)”. Does this imply "only 17.4% who delivered vaginally had previous caesarean section"?

Table 2: Compared to the data on text, it appears that there were more maternal deaths in caesarean sections than NVDs. Please confirm and correct.

Civil status of “united”, the phrase "united" may be easily understood in your country. However, the manuscript is written for distribution worldwide. Consider using a term that is easily understandable and that won’t be ambiguous in other regions.

Table 2: What does Antenatal Control mean?

Table 2: Death site: with regards to private institutions, the current results and percentages on this table gives the impression that deaths in private sector were far less. However, this was not the case as they are being shown as percentage of whole. When you calculate private hospitals alone, the picture is difference. For example, 140 of 341 is 41%, etc. Please rephrase.

The sentence that reads: “… terms of the place of death, it was found that 82.4% of the deaths of women who underwent a cesarean section occurred in a public institution, while in vaginal births, it was observed in 69.1% of cases.” Kindly rephrase this sentence to restore meaning.

Page 15: Sentence “Based on the data, it can be inferred that private institutions (p<0.001, CI 95%: 0.54-0.65) exhibited the highest tendency to perform this surgical procedure in women who eventually died.” Please review the percentage calculated as individual institution to make a better comparison.

Discussion

Reference this section: “The stratification into Robson's groups can be beneficial in identifying the obstetric and gynecological characteristics that have the greatest impact on the practice of cesarean sections. This classification system helps in distinguishing cases where the procedure was appropriate indicated from cases where it may have been unnecessary.”

Paragraph 4: IIn group 4 (multiparous women without a previous uterine scar, carrying a single pregnancy in cephalic position, at or beyond 37 WOG), a high proportion of cesarean sections (67.21%) was observed…” Is there no association with hypertensive diseases in this group as there was increased C/.S rate in hypertensives?. (they could be nulliparous, singleton, no previous C/S but abruption or f/d from insufficiency)

The sentence “Therefore, the high rate this group also suggests a potential overuse of this surgical intervention, disregarding the favorable likelihood of spontaneous labor for multiparous women.” Without knowing the indications for caesarean sections in all these categories, it is unfair to use the phrase "overuse of this procedure”.

The sentence “However, recent studies indicate that a previous cesarean is not an absolute indication for performing a cesarean section”, similarly, finding or describing the indications would be most relevant. I suggest you describe the indications first before making this strong statement.

The sentence “These findings further support the notion of an overuse of cesarean sections in Mexico, as these groups do not have inherent characteristics that necessitate a cesarean deliver”, what was reported as indications? This statement is unjust and unfair unless indications are known.

The sentence “It is interesting to note the disparity between the findings of the present study and the study conducted by Tura et al, where they reported that the largest proportion of cesarean sections was performed in groups 3, 5 and 1, with 21.4%, 21.1%, and 19.3% respectively” The main disparity is that your study was for patients who died. Therefore, it should not surprise you. These groups are not the same and cannot be compared directly.

The sentence “This discrepancy may indicate that, despite receiving adequate prenatal care, some cesarean sections are performed without sufficient medical justification during labor and delivery.” What were the recorded indications for caesarean section?

The sentence “This observation raises concerns that a cesarean section without clear clinical justification may increase the probability of maternal death.” This statement appears to be incorrect under the context.

The sentence “For instance, a study by Lisonkova et al. found that the risk of severe maternal morbidity was higher in planned cesarean deliveries compared…”, This reference looked at morbidity and not primarily mortality and therefore may not be compared directly. It was also predominantly on obese patients.

The sentence “The absence of a control group of surviving women significantly hampers the ability to ascertain causal relationships between cesarean section rates and maternal mortality…”, Was it not possible to have this control group? These patients’ records are likely to be available as well.

Conclusion

The conclusion about "overuse of caesarean sections" is not appropriate because the recorded indications at booking were disregarded during analysis of this data.

The stark contrast in C/S in private versus public sector warrant a comment about further exploration into the drivers of caesarean section in the private sector in your country or comment with available local data. This is probably not much different from other countries.

Reviewer #2: Results section, page 8 first paragraph needs to be reviewed - grammar errors

Table 2 - antenatal/prenatal control sounds weird, change wording

Cesarean section and c-section interchanged a lot throughout the article, sometimes in the same paragraph - please review

6. PLOS authors have the option to publish the peer review history of their article (what does this mean?). If published, this will include your full peer review and any attached files.

Reviewer #1: No

Reviewer #2: No

---

## [Author Response · Author response to Decision Letter 0]

19 Feb 2024

Cuernavaca, Mexico, February 16th 2023

Dr. Lawrence Chauke

Academic Editor

PLOS One

Dear Dr Chauke,

We are pleased to submit the revised version of our paper originally titled “Cesarean sections among maternal deaths in Mexico: Obstetric Risk Profiles and Other Correlates” (manuscript ID PONE-D-23-37685) for your consideration. As explained below, the title has changed to “Obstetric Risk Profiles and Causes of Death: Estimating their Association with Cesarean Sections Among Maternal Deaths in Mexico”. 

We sincerely appreciate your input and the reviewers’ constructive criticism. We believe this has greatly enhanced and improved the presentation of our work. 

Below, we respond point-by-point to specific journal requirements, your own comments and those of the reviewers. Comments appear in Italics.

Best regards,

Dr. Jimena Fritz

Corresponding Author

Journal Requirements:

We have ensured that the re-submission complies with PLOS ONE style requirements. 

We will share the code upon acceptance of the article. In the meantime we have uploaded it to FigShare (https://figshare.com) with restrictions. 

Thank you for the suggestion. We have uploaded the minimal dataset and the code to FigShare, with restrictions that will be lifted upon acceptance of the article. 

4. In the online submission form, you indicated that your data will be submitted to a repository upon acceptance. We strongly recommend all authors deposit their data before acceptance, as the process can be lengthy and hold up publication timelines. Please note that, though access restrictions are acceptable now, your entire minimal dataset will need to be made freely accessible if your manuscript is accepted for publication. This policy applies to all data except where public deposition would breach compliance with the protocol approved by your research ethics board. If you are unable to adhere to our open data policy, please kindly revise your statement to explain your reasoning and we will seek the editor's input on an exemption. 

Thank you for the suggestion. We have uploaded the minimal dataset and the code to FigShare, with restrictions that will be lifted upon acceptance of the article. 

The full ethics statement has been included in the revised version, at the beginning of the Methods Section, as requested. 

6. Please ensure that you refer to Figure 1 in your text as, if accepted, production will need this reference to link the reader to the figure.

We apologize for this omission and have included a call to Figure 1 in the revised version. 

Academic Editor Comments to Author:

Additional Editor Comments:

Dear Fritz, Jimena and colleagues

Thank you very much to you and your team for submitting your research for consideration for publication to PLOSONE. The paper addresses and important topic of global interest. The rising rate of caesarean section and the less than optimal decline in maternal mortality is a major concern globally . Indeed existing data suggests that the world might not achieve the Sustainable Development Goal (SDG) of reducing maternal mortality to less than 70 000 per 100 000 livebirths by the year 2030 if the situation continue as is. Studies like yours is important in that they provide additional insight into some of the factors that should be looked into in the global pursuit to achieve the 2030 SDG 3.1. You provide a compelling background and justification for the study . Thank you for doing that. The paper is a bit difficult to follow as is at present and perhaps you should look at the following points to try and improve the readability of your work and also the chances of the paper being published:

Thank you for your kind comments, we tried our best to produce an improved version of the manuscript, incorporating your valuable suggestions. As the additional analyses requested by you and the reviewers were extensive, we decided to invite an additional co-author with both clinical and research experience, who focused both on information retrieval and interpretation of results concerning immediate causes of death, co-morbid conditions, and indications for C-sections. Please note that in our original submission, we incorrectly stated that we analyzed deaths from 2010 to 2015. Actual deaths analyzed are from the 2010 to 2014 period, with the confusion arising from the year when data was first accessed (2015). We apologize for this mistake, which has been corrected in the revised version. 

1. Title

On reading the paper one gets the impression that the aim of the paper was to assess the association between mode of delivery, obstetrics risk factors and maternal deaths. If that is the case, you might want to revise the title to reflect this. Please refer to the PLOSONE author resource (https://plos.org/resource/how-to-write-a-great-title/) for guidance on how to write a winning title.

We appreciate your observation and suggest this alternative title: “Obstetric Risk Profiles and Causes of Death: Estimating their Association with Cesarean Sections Among Maternal Deaths in Mexico”. 

2.Research aim/objectives

Consider using scientific language to communicate the aim of the study and avoid using flowery words like unravel and elucidating. The reader should be very clear from reading the aim of the study as to what the study is all about. This is a bit not clear at present. May be the aim should be to assesses the association between the mode of delivery , maternal characteristics and maternal deaths or assess the contribution of mode of delivery and maternal characteristic on maternal deaths , for example.

We have tried to clarify the objective of the work in the revised version, while trying to maintain a more sober writing style over the abstract and main text.

3. Method

3.1. Thanks for describing the sources of your data

Thank you for your kind comments.

3.2. Consider writing this section in a past tense as recommended by the reviewers

We have ensured that the revised version of the methods is written in past tense. 

3.3. Outcome variables: It is very important to chose the outcome variables carefully to ensure that they answer the research question/ objectives. The socio-demographic and pregnancy variables chosen are appropriate. It would have been more useful to also look at co-morbid conditions as these could potentially influence maternal deaths. In other words, is it simply having a caesarean section or having a caesarean section in the background of other co-morbid conditions (or other factors) that increases the risk of maternal death?.

Thank you for your kind suggestion. We have re-examined the data and ascertained the presence of co-morbid conditions, specifically NCDs, as described in the updated Methods section. We found that the presence of co-morbidities (present in about 12% of women) was not significantly different between delivery modes. These results have been added to the revised paper, and are featured in Table 2. Evaluating whether these were related to risk of death, which is likely, is however outside our reach, as we lack a comparison group of women who did not die. 

 What about the experience of the surgeon and the level of care of the public healthcare facilities since majority of the deaths took place in public sector facilities. The following article might be of assistance ( Gebhardt G S, Fawcus S, Moodley J, Farina Z. Maternal death and caesarean section in South Africa: Results from the 2011 - 2013 Saving Mothers Report of the National Committee for Confidential Enquiries into Maternal Deaths. SAMJ, S. Afr. med. j. [Internet]. 2015 Apr [cited 2023 Dec 29] ; 105( 4 ): 287-291. Available from:

http://www.scielo.org.za/scielo.php?script=sci_arttext&pid=S0256-95742015000300025&lng=en. http://dx.doi.org/10.7196/SAMJ.9351.)

Thank you for your comment and the really interesting paper that you recommend us. As we know, since we have visited many health facilities in Mexico , , surgeons have a lot of work to do in the public sector, since the volume of patients is really high. That is one of the reasons a lot of gynecologists prefer to perform a CS section instead of waiting for hours for a vaginal and spontaneous delivery. Since they do not have a lot of health personnel nor midwifes inside the hospitals, they do not have plenty of time if an emergency occurs or if any pregnant women suddenly complicate. 

4. Results

4.1. From the data, it seems like the were more deaths in the normal vaginal delivery group compared to caesarean section but you seem to lean more on caesarean section related maternal deaths which are less compared to the normal delivery group.

Yes, but the 47% contrasts with 43% of all births being by C-section in 2010 , suggesting (even though this study can not prove it) c-sections might be playing a role. In addition, newly retrieved data now featured in the paper shows that a large number of c-sections appeared to be unjustified, see response to reviewer 1’s 6th comment. We have added Table 5 with data about the indications of c-sections in these women, adding results and discussion of these findings to the revised version.

 There is also notable significant difference between the women who died after caesarean section and those following normal delivery but this data does not seem to be addressed by the paper. It might be worth spending some time looking and scrutinising this data as this might be something you should look include in your multimodal regression analysis.

Most of the variables which showed statistical significance in the bivariate analysis were included in the logistic regression analysis, with the exception of “Delivery Care” as it is almost perfectly correlated with the performance of C-section (i.e. all C-sections are performed by doctors). Regardless, we have expanded the discussion to comment on these significant differences, in the revised version. 

4.2. Thank you for using the Robson classification to show the most indication for caesarean section and the risk of dying according to the Robson group.

Thank you for your kind comments. 

4.3. Consider separating diagnosis and cause of death. Haemorrhage is the triggering event but what did the women die of ? Is it hypovolaemic shock? Those with pre-eclampsia did they die from bleeding , brain haemorrhage , embolism, etc? You tried to do this in Table 4 but can be further improved. Perhaps you need to scrutinise the data even further.

We added two supplementary tables detailing the most frequently listed immediate causes of death, by maternal cause. We now discuss about the immediate causes of death in the revised version. 

5. Conclusion: It would be appropriate to comment on this after the revised version because of the lack of alignment between the aim of the study, challenges with the interpretation and the conclusion arrived at.

With the permission of the editor, we would like to keep our original conclusions, which we believe are now actually supported by the new findings regarding the specific indications of c-sections, as well as a more clearly expressed objective of the study-

6. Additional comments: Please see incorporate the specific comments from the reviewers in addition to the above.

We have done that to the best of our ability, and believe that the reviewer comments allowed us to add another layer of depth to this work.

I hope that the review does not discourage you. Looking forward to the revised version.

On the contrary, we are motivated by your insightful comments as well as those of the reviewers. We are happy to share the revised version and are convinced that our work has been greatly enhanced. Thank you for your consideration.

Reviewers' Comments to Author:

Reviewer:1

• Reviewer #1: General

Thank you for submitting the manuscript and the intention to share clinical experience and possible solutions for the Mexican populations with the rest of readership of PLOS ONE journals.

We thank the reviewer for his/her kind comments. 

• The title of the study “Cesarean sections among maternal deaths in Mexico: Obstetric Risk Profiles and Other Correlates” suggest that maternal death will be a major point of discussion. However, this was not the case. Perhaps consider revising the title.

We have expanded our characterization of maternal deaths, including additional details on immediate causes of death. Additionally, we have revised the title and believe the revised version more accurately reflects the aims of our work. 

• The manuscript requires Proofreading for scientific language and grammatical errors.

We have tried to improve the quality of the written language in the revised version of the manuscript. 

Introduction.

• Third paragraph about the trend of slowing in the decrease of maternal deaths… Please such statistics should be made available here for the reader to understand that there has been a slowdown.

According to the Global Burden of Disease, 2019, the MMR in Mexico decreased 15% from 1990 to 2000 (71.4 to 61.7 deaths per 100,000 live births); 14 % from 2000 to 2010 (61.7 to 52.9) but only 8% from 2010 to 2019 (52.9 to 48.6). This has been added to the introduction. 

• Fourth paragraph about potential strategies…This should be noted that it cannot lead to understanding the cause as association does not equal casualty.

The reviewer is absolutely right and we have always emphasized that no causal claims can be made based on these results. This is explicitly stated in the second sentence of said paragraph.

• The last paragraph rationalizing the study….

How was the deduction of the caesarean sections being unnecessary and potentially contributing to maternal deaths made?

Although in the original version of our manuscript, we relied on Robson categories to presume some c-sections might have been unnecessary, newly retrieved data from clinical summaries presented in the revised version, show that substantial numbers of the performed c-sections were inadequately indicated. We do not intend to conclude that in any of these particular cases, the unnecessary c-section played a role in the woman’s death, but, as states in the last paragraph of the introduction, these results add to the broader discussion on the topic. 

Study design, population, and data sources

• This section should 

---

## [Decision Letter · Decision Letter 1]

3 Apr 2024

Obstetric Risk Profiles and Causes of Death: Estimating their Association with Cesarean Sections Among Maternal Deaths in Mexico

PONE-D-23-37685R1

Dear Dr. Dear Fritz, Jimena and colleagues,

We’re pleased to inform you that your manuscript has been judged scientifically suitable for publication and will be formally accepted for publication once it meets all outstanding technical requirements.

Kind regards,

Hlengani Lawrence Chauke, MBCHB, BTh, Dip HIV Man, FCOG, MMED (O &G), MSc

Academic Editor

PLOS ONE

Additional Editor Comments:

Thank you for submitting the revised manuscript on maternal deaths associated with caesarian sections in Mexico. The purpose of this study was to analyze maternal deaths, focusing on obstetric risk factors, causes of death, and their association with caesarian sections. The findings provide valuable insights into the maternal deaths related to caesarian sections and emphasize the need for increased attention to the indications and circumstances surrounding these procedures. The article is well-written, with a solid introduction, method, and results sections. The statistical methods used are appropriate, and the discussion section exhibits a thorough understanding of the subject matter and relevant literature. The references are appropriately cited. In conclusion, the revised manuscript has addressed all the recommendations from the reviewers. Thank you for taking the time to incorporate the reviewers' recommendations.

Reviewers' comments:

Reviewer's Responses to Questions

**Comments to the Author**

1. If the authors have adequately addressed your comments raised in a previous round of review and you feel that this manuscript is now acceptable for publication, you may indicate that here to bypass the “Comments to the Author” section, enter your conflict of interest statement in the “Confidential to Editor” section, and submit your "Accept" recommendation.

Reviewer #1: All comments have been addressed

Reviewer #2: All comments have been addressed

2. Is the manuscript technically sound, and do the data support the conclusions?

Reviewer #1: Yes

Reviewer #2: Yes

3. Has the statistical analysis been performed appropriately and rigorously? 

Reviewer #1: Yes

Reviewer #2: Yes

4. Have the authors made all data underlying the findings in their manuscript fully available?

Reviewer #1: Yes

Reviewer #2: Yes

5. Is the manuscript presented in an intelligible fashion and written in standard English?

Reviewer #1: No

Reviewer #2: Yes

6. Review Comments to the Author

Reviewer #1: Thank you for addressing all comments. The manuscript is well written, results are well presented, and the modified discussion chapter shows marked improvements.

Kindly subject this paper to Proof Reading either by a Professional Proofreader or utilize the services from PLOS ONE.

Reviewer #2: (No Response)

7. PLOS authors have the option to publish the peer review history of their article (what does this mean?). If published, this will include your full peer review and any attached files.

Reviewer #1: **Yes: **Dr. Langanani Mbodi

Reviewer #2: No

---

## [Editor Report · Acceptance letter]

26 Apr 2024

PONE-D-23-37685R1 

PLOS ONE

Dear Dr. Fritz, 

I'm pleased to inform you that your manuscript has been deemed suitable for publication in PLOS ONE. Congratulations! Your manuscript is now being handed over to our production team.

Kind regards, 

on behalf of

Prof Hlengani Lawrence Chauke 

Academic Editor

PLOS ONE